# Hyperthermic Intraperitoneal Chemotherapy in Ovarian Cancer

**DOI:** 10.3390/diagnostics10010043

**Published:** 2020-01-14

**Authors:** McKayla J. Riggs, Prakash K. Pandalai, Joseph Kim, Charles S. Dietrich

**Affiliations:** 1Division of Gynecologic Oncology, University of Kentucky, Lexington, KY 40502, USA; mbri229@uky.edu; 2Division of Surgical Oncology, University of Kentucky, Lexington, KY 40502, USA; prakash.pandalai@uky.edu (P.K.P.); joseph.kim@uky.edu (J.K.)

**Keywords:** HIPEC, ovarian cancer, peritoneal cancer, review, technique

## Abstract

Hyperthermic intraperitoneal chemotherapy (HIPEC) in conjunction with cytoreductive surgery (CRS) holds promise as an adjunctive treatment strategy in malignancies affecting the peritoneal surface, effectively targeting remaining microscopic residual tumor. HIPEC increases concentrations of chemotherapy directly within the peritoneal cavity compared with the intravenous route and reduces the systemic side effects associated with prolonged adjuvant intraperitoneal exposure. Furthermore, hyperthermia increases tissue penetration and is synergistic with the therapeutic chemotherapy agents used. In ovarian cancer, evidence is building for its use in both primary and recurrent scenarios. In this review, we examine the history of HIPEC, the techniques used, and the available data guiding its use in primary and recurrent ovarian cancer.

## 1. Introduction

First explored in ovarian malignancies in the 1980s, hyperthermic intraperitoneal chemotherapy (HIPEC) in conjunction with cytoreductive surgery (CRS) has become central in controlling peritoneal disease and improving survival outcomes in selected patients with peritoneal surface malignancies [1]. Peritoneal carcinomatosis was historically seen as the final and fatal stage in malignancy, and only in the past twenty years has an effective treatment evolved [2]. In addition to advanced-stage ovarian cancer, approximately 10–15% of patients with primary gastrointestinal malignancies (including appendiceal, colonic, and gastric malignancies) present with peritoneal carcinomatosis [3]. The goal of HIPEC is to target remaining microscopic residual tumor at the time of cytoreductive surgery following an optimal resection. The rationale of HIPEC compared with standard systemic chemotherapy is multifold. By delivering chemotherapy directly within the peritoneal cavity, increased exposure of poorly vascularized tumor to the high concentration of chemotherapy occurs. The blood–peritoneal barrier also limits the passage of this high concentration of chemotherapy, minimizing systemic toxicity while maximizing local effects. Heat is directly cytotoxic, improves chemotherapy penetration into tissue, and is synergistic with commonly used chemotherapy agents including cisplatin, paclitaxel, oxaliplatin, and mitomycin c. Hyperthermia has been shown to reduce the mechanisms of induced cellular resistance to cisplatin [4]. In both noninvasive and invasive peritoneal surface malignancies, the completeness of cytoreduction score (CC-) is the major prognostic indicator [5]. Candidates that receive the most benefit are those who undergo complete cytoreduction, although patients with small-volume residual disease are also eligible. In this review, we aim to examine the history of HIPEC, describe the techniques used, and explore its application in primary and recurrent ovarian cancer.

## 2. Evolution of HIPEC

Exploration into intraperitoneal versus intravenous chemotherapy began in the late 1970s. Initial studies explored cisplatin use in canines and found the levels of drug in tissues of the intraperitoneal cohort were 2.5–8 times higher than the intravenous group even four days following administration [6]. Further evidence suggested that the pharmacokinetics of intraperitoneal chemotherapy was improved by maintaining a higher concentration in the peritoneal space than in the plasma due to lower peritoneal permeability [7,8]. At the same time, research groups investigated the effect of hyperthermia on cancer cell destruction, where 42–43 °C was found to eradicate tumor in vivo [9]. Hyperthermia as a treatment modality was first described in 1893 when Coley reviewed the cases of 38 patients with high fevers and cancer, with 12 patients having complete regression of tumors and 19 patients improving in condition [10]. Early use of total body hyperthermia via hot baths was described in inoperable cervical cancer patients by Westermark in 1898 and penile carcinoma patients by Goetze in 1939 [11,12]. Both regional and total body hyperthermia were a focus of early study, primarily investigating the effect of hyperthermia on radiation efficacy in the 1910s–1950s [9]. Regional hyperthermia was examined using ultrasound and microwave antennas to localize the focus of intensity. In a study by Luk in 1980, he and his colleagues reported that tumor regression was more likely when radiation and hyperthermia were combined compared with either modality alone. In 1979, the first human subject underwent HIPEC for persistent pseudomyxoma peritonei that had failed laparotomy with cytoreduction. This patient received thiotepa followed by methotrexate, survived the operation, and was discharged without complications [13]. Over the next decade, multiple phase I and II trials attempted to determine the best techniques and practice for utilizing HIPEC.

In 1987, a phase I trial established the pharmacokinetic advantage of intraperitoneal chemotherapy over the intravenous route with cisplatin and etoposide, catching the attention of the clinical community [14,15]. Since ovarian cancer primarily metastasizes within the peritoneal cavity, intraperitoneal (IP) chemotherapy emerged as a strategy for dose intensification and to treat disease sites without a robust vascular supply. Gynecology Oncology Group (GOG) Study 104 and 114 showed significant improvements in overall survival in the IP arms, but these trials were largely discounted as concurrent survival gains with less toxicity were seen as paclitaxel and carboplatin were introduced into standard practice [16,17]. In 2006, the National Cancer Institute (NCI) issued a rare clinical alert after the results of GOG 172 were made available. A total of 415 patients that had an optimal cytoreduction with no residual mass greater than 1.0 cm were randomly assigned to two arms. In the first arm, patients received intravenous paclitaxel (135 mg/m^2^) over a 24 h period, followed by intravenous cisplatin (75 mg/m^2^) on day 2. In the second arm, patients received intravenous paclitaxel (135 mg/m^2^) on day 1, intraperitoneal cisplatin (100 mg/m^2^) on day 2, and intraperitoneal paclitaxel (60 mg/m^2^) on day 8. All patients received treatment every three weeks for six cycles. Progression-free survival (PFS) was improved by 5 months in the IP arm (18.3 vs. 23.8 months, *p* = 0.05) and overall survival by nearly 16 months (49.7 vs. 65.6 months, *p* = 0.03) [18]. Long-term analysis of 876 patients from GOG 114 and GOG 172 showed the survival advantage of IP chemotherapy over IV chemotherapy extends beyond 10 years and the risk of death decreased by 12% for each cycle of IP chemotherapy completed [19]. Despite the dramatic improvement in overall survival, adoption of IP chemotherapy in the community setting has been low. Only 42% of patients in GOG 172 were able to complete six cycles of therapy with a significantly lower quality of life during treatment, and administration of IP chemotherapy can be logistically difficult [18]. Others have pointed out that the IP arm in GOG 172 was really a dose-dense paclitaxel regimen and was compared with the wrong control. GOG 252 was developed to address these concerns. The control arm used intravenous dose-dense weekly paclitaxel and carboplatin. Arm 2 received intravenous dose-dense paclitaxel and IP carboplatin on day 1. Arm 3 was a modified GOG 172 regimen with a three-hour infusion of intravenous paclitaxel on day 1, a lower IP cisplatin dose (75 mg/m^2^) on day 2, and the same IP paclitaxel dose on day 8. All arms also received bevacizumab during cycles two through six and then as maintenance through cycle 22. There were no differences in progression-free survival (PFS) or overall survival (OS) outcomes in the three arms [20]. It is anticipated that with the publication of this trial, IP chemotherapy will be less utilized going forward due to decreased toxicity and improved patient-reported outcomes with IV chemotherapy without a compromise in PFS or OS. However, the addition of bevacizumab in GOG 252 may have masked the true benefit of IP chemotherapy.

Despite the theoretical advantages of IP chemotherapy and survival benefits reported in GOG 172, significant challenges in delivering treatment have limited its use in practice. Application of HIPEC to the ovarian cancer population has renewed interest due to its advantages over standard adjuvant IP protocols. HIPEC is a single treatment delivered at the time of cytoreductive surgery, where fluid distribution is not impeded by postoperative adhesions and the surgeon can control delivery time and distribution using standardized techniques. Furthermore, patient tolerability has been excellent [21].

## 3. Technique

The following is a description of our technique for cytoreduction and HIPEC in peritoneal surface malignancies. This procedure is a concordant effort between surgical oncology and gynecologic oncology at our institution.

## 4. Cytoreduction

Eradication of all gross and microscopic disease by surgical resection and heated chemotherapy are the principles of CRS and HIPEC. This technique is ideally suited for pathology with peritoneal surface involvement without visceral or extra-abdominal metastases [22]. CRS/HIPEC is typically performed through a generous midline laparotomy incision from xiphoid to pubis symphysis. Care is taken to avoid iatrogenic injury during entry into the peritoneal cavity as many patients have had prior surgery as well as the adhesions that are associated with peritoneal carcinomatosis. The falciform ligament is taken between large clamps or using an energy device to facilitate exposure of the liver and diaphragm. A Thompson retractor allows for wide exposure. If CRS/HIPEC is being performed for pseudomyxoma peritonei, surgeons usually encounter mucin, which is typically removed manually or with suction catheters. Tenacious, thick mucin may require manual removal using laparotomy pads. The peritoneal carcinoma index (PCI) standardizes the initial quantification of the disease burden, separating the abdominal cavity into 13 distinct regions. The surgeon assigns tumor burden scores for each of these regions based upon the tumor volume in each region [23]. Maximal cytoreduction is critical for survival benefit [24]. Ominous signs of aggressive biology are large implants within the root of the small bowel mesentery, unrecognized liver metastases suggestive of systemic spread, or widespread serosal implants on the small bowel.

Preoperative imaging with contrast-enhanced CT has reasonable sensitivity in detection of solid organ involvement, while MRI has greater sensitivity in detection of peritoneal surface disease. Neither CT nor MRI reliably defines low-volume involvement of the small bowel mesentery, which tends to be difficult to fully eradicate and is associated with worse outcomes [25]. After initial exploration to quantify PCI and rule out unrecognized solid organ metastases, we proceed with resection of all sites of gross disease. Regardless of detection of gross disease, we advocate standard resection of the greater and lesser omentum, appendix, gallbladder, and in women, a total abdominal hysterectomy and bilateral salpingo-oophorectomy. We use a thermal energy device (Ligasure, Covidien, Minneapolis, MN, USA) to remove the omentum by dividing the gastrocolic ligament close to the transverse colon from the hepatic to the splenic flexures. The omentum is then divided along the greater curvature of the stomach while preserving the gastro-epiploic vessels. The dissection is carried up along the greater curvature to the short gastric vessels and across to the hilum of the spleen, where the omentum is transected completely. The appendix is usually mobilized, and the mesoappendix is controlled with either Ligasure or electrocautery. The base is transected with a linear stapling device. Dome-down cholecystectomy is performed with identification of the cystic artery and duct, and titanium clips placed on these two critical structures prior to completing cholecystectomy. This dissection may be difficult as mucin and carcinomatosis tends to conglomerate or pool in the porta-hepatis as an entrance point through the Foramen of Winslow to the lesser space. Many patients will also require stripping of the right and occasionally left hemi-diaphragm. We utilize long Allis clamps to grasp the peritoneal lining of the diaphragm and “strip” only gross disease. If the thoracic cavity is entered, primary closure is usually performed around a red rubber catheter. A purse string suture is placed around the catheter prior to final closure. Anesthesia is instructed to give a Valsalva and hold while a closed suction is used to evacuate air from the thoracic space as the purse string is secured. This combination is usually enough to avoid intraoperative pneumothorax and chest tube placement. Another common site of disease is on the liver surface. We utilize the bipolar hemostatic device (Aquamantys, Medtronic, Minneapolis, MN, USA) to strip Glisson’s capsule of the liver. Aquamantys can also be utilized for small volume deposits on the mesentery of the small bowel or on the surface of the spleen. While generally reserved for low-grade malignancy, the thick tenacious nature of mucinous disease may coat the pelvis, rectum, lesser curvature of the stomach, antrum of the stomach, and porta-hepatis. It is common to perform major multivisceral resection to completely remove disease.

## 5. Heated Intraperitoneal Chemotherapy Administration

The rationale for using HIPEC is to eradicate all microscopic disease with higher local concentrations of cytotoxic chemotherapy further potentiated with heat. Following resection of all gross disease, the abdomen is washed with three liters of normal saline to remove any small fat or particulate that may cause clogging of the out-flow circuit in preparation for HIPEC. Enteric reconstruction is delayed until after perfusion is completed to avoid exposure of the anastomosis to the peritoneal dosage of chemotherapy. Mitomycin C or oxaliplatin are the agents of choice for colorectal and appendiceal tumors, while cisplatin is used for gastric cancer and gynecologic malignancy. The American Society of Peritoneal Surface Malignancies (ASPSM) has advocated the use of Mitomycin C for patients with peritoneal surface disease of colorectal origin [26]. In gynecologic patients with platinum-resistant disease, either paclitaxel or Mitomycin C can be considered.

HIPEC is performed using an open platform (referred to as the coliseum technique) or a closed system, which is our preferred method (Figure 1 and Figure 2). The coliseum technique involves sewing a large nonporous synthetic mesh to the skin edges and tenting the abdomen up using the Thompson retractor to create a tent or coliseum. The surgeon manually stirs the chemotherapy perfusate to ensure equal distribution throughout the abdominal cavity. Due to concerns about exposure and spillage, the open technique has been abandoned in some institutions. We employ the Genesis Medical System (Figure 3), which utilizes two inflow catheters connected via a Y connector to a single primary catheter and a single outflow “lasso” catheter. In our experience, this system rapidly achieves goal temperature once catheters are placed and skin is closed tightly with a running large nylon suture. The abdomen is warmed with hyperthermic saline to 42–43 °C. Once the target temperature is reached, chemotherapy is infused into the circuit and perfused for 30–90 min with simultaneous vigorous manual agitation of the abdomen. After perfusion is completed, the heated chemotherapy solution is drained from the abdomen via outflow catheters, three liters of saline are washed thru the circuit, and the skin is carefully reopened. Once retractors have been placed, an additional three liter wash is performed, and the final reconstruction is performed or ostomies constructed.

## 6. Applications in Primary Ovarian Cancer

Several case series and case-control studies established the feasibility and safety of HIPEC in ovarian cancer patients. Gori et al. published a case-control series in 2005 using cisplatin 100 mg/m^2^ for HIPEC consolidation after treatment with IV chemotherapy. The treatments were well tolerated, but no significant differences in 5 year survival were demonstrated [27]. Kim et al. also reported on 18 patients with Stage Ic–IIIc ovarian cancer with negative second-looks that received consolidation HIPEC with paclitaxel at the time of laparotomy and compared them with 24 patients receiving conventional therapy. HIPEC was well tolerated with no significant trends towards increased morbidity. Both PFS and OS were better in the HIPEC group, although small numbers and the wide stage presentations limited this analysis [28]. Rettenmaier et al. reported on 37 patients who had a complete response to primary chemotherapy and then had consolidation HIPEC using carboplatin AUC 10 followed by paclitaxel maintenance. The treatment was well tolerated. Median disease-free survival was 13 months and overall survival was 14 months in this small retrospective series [29].

Multiple studies demonstrating the benefit of HIPEC in ovarian cancer have led to randomized clinical trials in the upfront setting. These trials are summarized in Table 1. Ansaloni et al. published an open, prospective phase II study in 2012 that enrolled 39 patients. Nine of these received HIPEC during their primary treatment. The overall mean recurrence time was 14.4 months, with improved outcomes reported in patients achieving complete cytoreduction [30]. Bakrin et al. published a French multicenter retrospective cohort in 2013. In their experience with HIPEC in 566 advanced ovarian carcinoma patients, 92 patients had treatment in the first-line setting and had a median overall survival of 35.4 months and median recurrence-free survival of 11.8 months. Twelve had HIPEC at first look, 24 at an interval surgery, and 56 for consolidation. In patients with CC-0, median survival was 41.5 months. Concerns with this study include lower survival rates when compared with historical studies and a higher risk of complications in the upfront setting. However, the population in this cohort was mainly patients referred to tertiary centers with a worse initial prognosis due to higher stage and often extensive unresectable disease [31]. Cascales-Campos et al. reported on 87 patients with Stage III/IV ovarian cancer treated at tertiary centers in Spain from 1998 to 2011. HIPEC was offered as part of the primary treatment beginning in 2008, and 52 patients received HIPEC in the upfront setting after complete cytoreduction. Of these patients, 44% had a primary debulking procedure and 56% had an interval procedure. Paclitaxel 60 mg/m^2^ at 42 °C with a dwell time of 60 min was used during the HIPEC portion. Disease-free survival was significantly improved at one and three years in patients receiving HIPEC (81% and 63% vs. 66% and 18%, *p* < 0.01) [32]. In a meta-analysis of 9 comparative studies and 28 cohort studies in primary epithelial ovarian cancer, Huo et al. showed that HIPEC + CRS + chemotherapy had significantly improved 2, 3, 4, 5, and 8 year overall survival when compared with CRS + chemotherapy alone [33].

The first multicenter phase III randomized controlled trial using HIPEC in the upfront setting in ovarian cancer was published by van Driel et al. in the New England Journal of Medicine in 2018 [34]. This landmark paper analyzed 245 patients with poor prognosis stage III ovarian cancer who had undergone neoadjuvant IV chemotherapy with carboplatin area under concentration (AUC) AUC 5 or 6 and paclitaxel 175 mg/m^2^ and had stable disease after three cycles. The patients were randomized at the time of surgery if an optimal cytoreduction was felt to be feasible to either cytoreduction with or without HIPEC using the GOG 172 dosage of cisplatin 100 mg/m^2^. The patients then received three additional cycles of adjuvant IV chemotherapy. The surgery plus HIPEC group had a 4 month progression-free survival advantage and a 12 month overall survival advantage. Of note, grade 3 or 4 adverse events were not significantly different between the groups. A primary criticism of this study is that their control arm had a progression-free survival markedly lower than the presumed 18 months for statistical analysis, which could explain the difference between the arms. However, even if a true difference existed, this improvement in progression-free survival is similar to the benefit seen with bevacizumab in front-line treatment of suboptimally debulked epithelial ovarian cancer. The study also focused primarily on overall survival data, though the primary endpoint was progression-free survival.

Preliminary results for a second randomized controlled trial were presented at the American Society of Clinical Oncology annual meeting in 2017 by Lim et al. In this Korean trial, 184 patients with stage III and IV epithelial ovarian cancer were randomized to receive HIPEC or no HIPEC after an optimal upfront or interval cytoreduction. Cisplatin 75 mg/m^2^ with a 90 min dwell time was used for patients undergoing HIPEC. No differences in postoperative complications were noted between groups. No differences in PFS at 2 years (43.2% vs. 43.5%) or 5 years (20.9% vs. 16.0%) were noted in the HIPEC versus control groups (*p* = 0.569). Five-year OS was also similar between groups (51.0% and 49.4%, *p* = 0.574). In the neoadjuvant chemotherapy subgroup, the PFS and OS curves after 20 and 30 months respectively showed a trend towards gradual distinction favoring HIPEC, allowing the authors to conclude that more long-term follow-up is needed to confirm the impact of HIPEC in this population [35]. Unlike the Dutch randomized trial which showed a significant benefit in Stage III patients having an interval procedure, this Korean trial pooled Stage III and IV patients and used HIPEC in both upfront and interval cytoreduction. They also used a lower cisplatin dose, a de-escalation strategy that similarly did not pan out in GOG 252.

Several key questions remain unanswered when using HIPEC in the primary treatment of ovarian cancer. The first revolves around the optimal timing of HIPEC. It has been successful during both upfront surgery and interval cytoreduction following neoadjuvant chemotherapy. It has also been explored as a consolidation strategy, although this application is not as appealing as second-look laparotomies are not considered standard of care any longer. Based on the randomized trials available, it appears that using it at interval cytoreduction holds the most promise, and the latest iteration of the National Comprehensive Cancer Network (NCCN) Guidelines supports this approach [36]. OVHIPEC-2 (NCT03772028) has been designed and will hopefully answer whether HIPEC during upfront cytoreduction is also beneficial [37]. Further questions regarding the most appropriate chemotherapeutic agent, dosing, dwell time, and temperature also exist. Cisplatin 100 mg/m^2^ is recommended in the NCCN Guidelines, although paclitaxel is another reasonable option that has not been evaluated in a randomized controlled fashion. De-escalation of cisplatin to 75 mg/m^2^ has not been advantageous in either the Korean HIPEC randomized controlled trial (RCT) nor in GOG 252. The last controversy regarding HIPEC, and perhaps the more difficult one to answer, centers on how it compares or fits in with other rapidly evolving strategies being used in the front-line setting, including the use of bevacizumab and Poly (ADP-ribose) polymerase (PARP) maintenance, adjuvant IP chemotherapy, and immunotherapy. Hopefully, future carefully designed clinical trials will address these knowledge deficits. Until then, it is reasonable to consider HIPEC in select patients at institutions experienced with the technique.

## 7. Applications in Recurrent Ovarian Cancer

Evidence for the benefit of HIPEC in recurrent ovarian cancer is currently limited to single institution experience and retrospective series. In a systematic literature review by Hotouras, et al., sixteen studies were described, ranging from level II-IV evidence [38]. No consensus currently exists for the specific chemotherapy agent to be used, protocol for delivering, and postoperative therapy. A French randomized controlled trial is currently underway, CHIPOR (NCT01376752), with an estimated primary completion in fall of 2024. The Italian HORSE (NCT01539785) trial completed in September 2018 explores the role of HIPEC in open CRS versus minimally invasive CRS. Several smaller level III trials and one level I trial provide preliminary support for the use of HIPEC in recurrent ovarian cancer with promising results. These findings are summarized in Table 2.

In 2012, Fagotti and colleagues performed a case-control study comparing 30 platinum-sensitive epithelial ovarian cancer patients who underwent secondary cytoreductive surgery (CRS) and HIPEC to 37 patients who had CRS without HIPEC [40]. Oxaliplatin 460 mg/m^2^ was used for 30 min at 41.5 °C with closed technique. Systemic platinum-based chemotherapy then followed for six cycles. The primary endpoint was progression-free survival and was found to be 26 months in the HIPEC group compared with 15 months in the non-HIPEC group. Recurrence occurred in 66.6% of the HIPEC group and 100% of the control group with mean follow up time of 46 months and 36 months respectively. No delay to starting adjuvant chemotherapy was found following HIPEC administration with an average time of 40 days. HIPEC toxicity occurred in 35% of patients with no mortality.

Bakrin et al. performed a multicenter prospective study of outcomes in 246 patients with recurrent epithelial ovarian cancer, both platinum-sensitive (*n* = 184) and platinum-resistant (*n* = 62), who had been treated with optimal CRS and HIPEC [41]. The HIPEC techniques used were both open using the coliseum procedure and closed with the Lyon device. CC scores were recorded, and HIPEC was performed at the time of operation following CRS. Cisplatin was used in 95.5% of procedures alone or in combination with doxorubicin or mitomycin C for 90 min. Inflow temperatures ranged from 44 to 46 °C. Mean age of patients was 57.5, mean body mass index (BMI) 23.7, and 92.2% of patients had a CC of zero or one defined as <2.5 mm residual tumor. No patients received bevacizumab during the study period. Grade III or IV complications occurred in 11.6% of the patients, including grade 3 leukopenia, hemorrhage, and postoperative anastomotic leakage. Overall median survival was 48.9 months, with 48 months in platinum-resistant disease and 52 months in platinum-sensitive disease (*p* = 0.568). Overall survival rates were 86%, 60%, and 35% at years 1, 3, and 5, respectively.

Cascales-Campos and colleagues examined both primary stage IIIc and recurrent ovarian cancer patients who underwent optimal cytoreduction to CC-0 or CC-1 with HIPEC and analyzed patient outcomes in the postoperative period [39]. Their HIPEC procedure utilized 60 mg/m^2^ of cytostatic paclitaxel, and 75 mg/m^2^ cisplatin in patients with anaphylaxis to paclitaxel or docetaxel previously, in a coliseum technique for 60 min. Temperature was kept constant between 42 and 43 °C. They used a fast tract protocol adopting many of the Enhanced Recovery After Surgery (ERAS) tenets. Median operative time was 380 min. CC-0 was achieved in 38 of 46 patients, and CC-1 in the remaining eight. Major morbidity was 15.3%, with paralytic ileus being the most common. Three patients developed pleural effusions, two had wound infections, and two self-limited postoperative bleeding cases were identified. Mean postoperative stay was 6.9 days (range 3–11 days). No mortality related to the procedure was found.

Safra and colleagues performed a case-control study in recurrent ovarian cancer patients and compared 27 treated with CRS and HIPEC to 84 matched patients in a 1:3 ratio who received systemic chemotherapy alone [42]. The HIPEC regimen included either cisplatin 50 mg/m^2^ plus doxorubicin 15 mg/m^2^ or paclitaxel 60 mg/m^2^ plus carboplatin AUC 4. Rarely, the patients received cisplatin 25 mg/l/m^2^ plus mitomycin-C 3.3 mg/l/m^2^. HIPEC time was 120 min with a target temperature of 42.5 °C. All patients recovered in the intensive care unit for at least 48 h postoperatively. No patients were treated with bevacizumab in the study. Median overall survival data had not yet matured at the time of publication, with over 70% of patients still alive at the time of analysis. Median progression-free survival was 15 months in the HIPEC group and 6 months in the systemic chemotherapy group (*p* = 0.001). A paired matched analysis was performed in BRCA gene mutation carriers, and a progression-free survival of 20.9 months with HIPEC compared with 12.6 months without HIPEC was noted (*p* = 0.048). Overall survival appeared comparable, though nearly 70% of the HIPEC patients were alive at the analysis compared with less than 40% of the control group.

In 2015, Spiliotis and colleagues published the first randomized controlled trial of recurrent ovarian cancer and HIPEC. They examined 120 women with stages IIIc–IV epithelial ovarian cancer who had recurred following prior treatment with cytoreductive surgery and systemic chemotherapy [45]. The patients were randomized into two treatment groups: secondary CRS with HIPEC plus subsequent systemic chemotherapy or secondary CRS plus subsequent systemic chemotherapy alone. The HIPEC protocol regimen varied depending on whether the patients were platinum-sensitive or platinum-resistant. Platinum-sensitive patients (*n* = 34) received cisplatin 100 mg/m^2^ and paclitaxel 175 mg/m^2^, delivered for 60 min at 42.5 °C. Platinum-resistant patients (*n* = 26) received doxorubicin 35 mg/m^2^ and either paclitaxel 175 mg/m^2^ or mitomycin 15 mg/m^2^ for 60 min at 42.5 °C. HIPEC was performed with open coliseum technique in forty patients, and closed technique in the remaining twenty. All procedures were completed by the same surgical team.

The primary study outcome was mean overall survival. Mean overall survival in the HIPEC group was 26.7 months compared with 13.4 months in the non-HIPEC group (*p* = 0.006). When examined by stage, stage IIIc with HIPEC had a survival of 26.9 months, and 14.2 months in the non-HIPEC group. This was also seen in the stage IV disease patients, with HIPEC survival at 26.4 months and non-HIPEC at 11.9 months. When comparing platinum-resistant with platinum-sensitive disease responsiveness, the survival of HIPEC patients with platinum-resistant stage IIIc (26.08 months) was not statistically different than for those with platinum-sensitive (27.28 months, *p* = 0.287). This was also seen in stage IV disease. The group also explored the importance of cytoreduction on overall survival using the CC score with a goal of no residual disease. In patients who received HIPEC with a CC-0, survival was 30.9 months. At a CC-1 level, survival dropped to 23.9 months, and CC-2 was even lower at 12.1 months. This was compared with the non-HIPEC group, with CC-0 being 16.1 months, CC-1 at 11 months, and CC-2 at 6.7 months. These were statistically different with *p* = 0.002. No data was provided regarding progression-free survival, information on what systemic chemotherapy was given afterwards, and no data regarding complications or morbidity. Also of note, no difference was noted between platinum-sensitive and platinum-resistant cohort overall survival irrespective of HIPEC administration, as would be expected.

In the recurrent setting, no consensus exists on the protocol to be used. This includes variation in the choice of intraperitoneal chemotherapy drug, length of time, timing with surgery, open versus closed technique, and temperature. The Spiliotis team utilized cisplatin and paclitaxel in platinum-sensitive disease, doxorubicin and paclitaxel or mitomycin in platinum-resistant disease, and followed with subsequent chemotherapy using single-agent therapy to minimize the cumulative toxicity [45]. Fagotti described the use of oxaliplatin [40,47]. Bakrin describes techniques using cisplatin, mitomycin, or doxorubicin [41]. Cascales-Campos describes using primarily paclitaxel [39]. Pharmacokinetics of cisplatin during cytoreduction surgery plus HIPEC in platinum-sensitive ovarian cancer was explored prospectively by Petrillo and colleagues in Italy, where they showed a benefit to minimally invasive routes with enhanced cisplatin peritoneal uptake during HIPEC [48].

Though the available literature on HIPEC in recurrent ovarian cancer seems overwhelmingly favorable, no prospective multicenter randomized controlled trials have been published to date to evaluate the true benefit and optimized protocol for HIPEC in recurrent settings. This is a critical area of future research given the promising preliminary data currently available.

## 8. Conclusions

Although the oncologic community awaits more evidence for the benefit of HIPEC use in ovarian cancer, several studies have shown its potential as a therapeutic adjunct; however, several questions remain. To date, most success with HIPEC has been seen when used during an interval procedure or in the recurrent setting, but ongoing trials are examining the most optimal timing. Other areas for future exploration include the best chemotherapy agent, a uniform procedure for HIPEC administration, and how to integrate HIPEC with subsequent therapy. Though criticisms are present in many of the trials, there appears to be a clear trend towards survival benefit with its use while maintaining a well-tolerated side effect profile. The future of HIPEC and its adoption into routine clinical practice will depend on the development of well-designed clinical trials to answer these important questions.

## Figures and Tables

**Figure 1 diagnostics-10-00043-f001:**
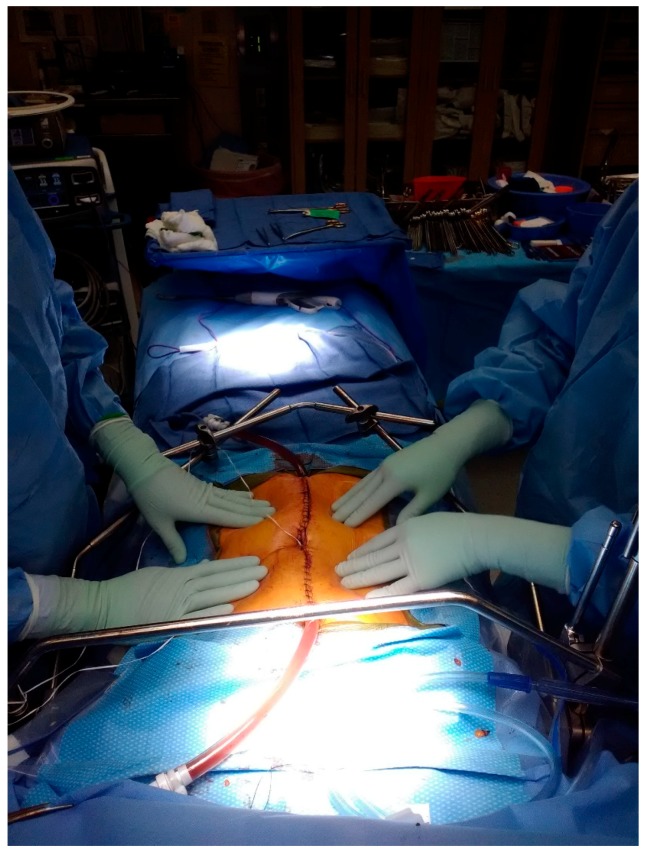
Closed-system hyperthermic intraperitoneal chemotherapy (HIPEC) technique.

**Figure 2 diagnostics-10-00043-f002:**
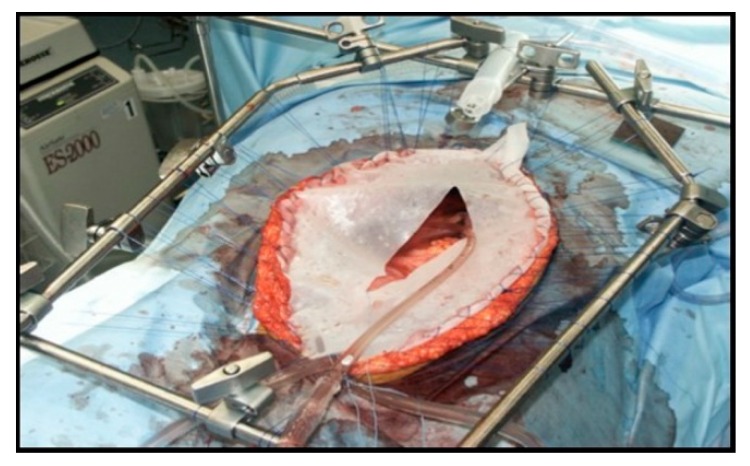
Coliseum HIPEC technique.

**Figure 3 diagnostics-10-00043-f003:**
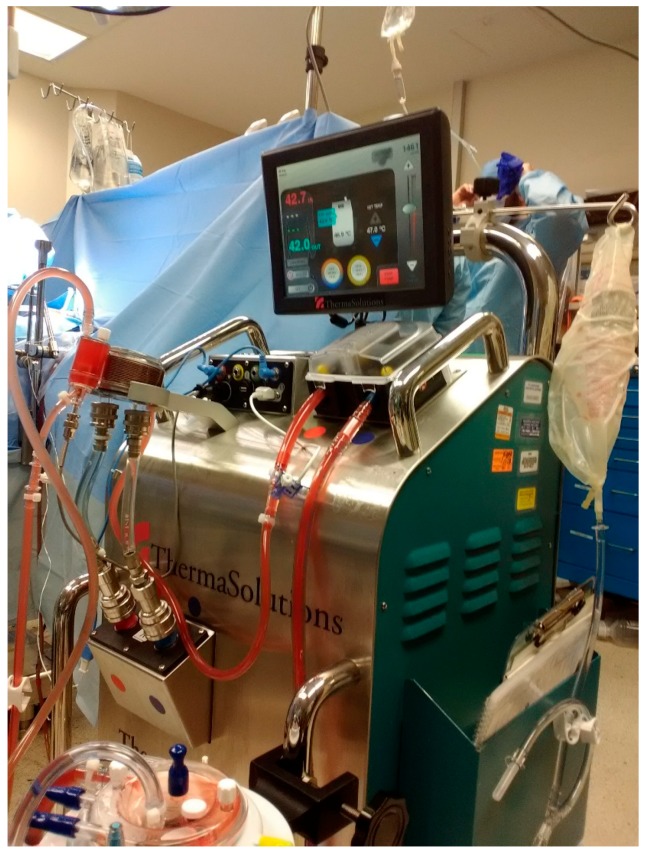
Genesis Medical System for hyperthermic chemotherapy.

**Table 1 diagnostics-10-00043-t001:** Review of intraperitoneal chemotherapy trials in primary ovarian cancer.

Study (Level of Evidence)	Cohorts	Drugs	Results
Ansaloni, 2012, Level II [30]	Open, prospective, nonrandomized phase II study in patients with primary or recurrent peritoneal carcinomatosis with ovarian cancer ● Primary (*n* = 9) ● Recurrent (*n* = 30)	● Cisplatin 100 mg/m^2^ ● Paclitaxel 175 mg/m^2^ ● Doxorubicin 35 mg/m^2^ ● 90 min at 41.5 °C Agent(s) determined based on prior systemic chemotherapy that had been received. 66% received cisplatin + doxorubicin.	● CC-0 (complete cytoreduction score-0) achieved in 90% ● Mean hospital stay 23.8 days ● One postoperative death ● No significant differences in 5 year survival in primary ovarian cancer cohort
Lim, 2017, Level I [35]	Randomized controlled trial in primary ovarian cancer staged III and IV ● HIPEC + Cytoreductive surgery (CRS) + Systemic chemotherapy (CT) (*n* = 92) ● CRS + Systemic CT (*n* = 92)	● Cisplatin 75 mg/m^2^ ● 90 min at 41.5 °C	5 year progression-free survival (*p* = 0.569) ● HIPEC: 20.9% ● Control: 16% Five year overall survival (*p* = 0.574) ● HIPEC: 51% ● Control: 49.4% No statistical difference in morbidities between groups
Van Driel, 2018, Level I [34]	Multicenter prospective randomized controlled phase III trial Stage III ovarian cancer who had received neoadjuvant IV chemotherapy with carboplatin and paclitaxel with stable disease after 3 cycles. ● CRS + HIPEC ● CRS alone	● Cisplatin 100 mg/m^2^ ● 90 min at 40 °C	Primary end point: progression-free survival ● HIPEC: 14.2 months ● Control: 10.7 months Median overall survival ● HIPEC: 45.7 months ● Control: 33.9 months Adverse events of grade 3 and 4: 25% and 27%

**Table 2 diagnostics-10-00043-t002:** Review of intraperitoneal chemotherapy trials in recurrent ovarian cancer.

Study (Level of Evidence)	Cohorts	Drugs	Results
Cascales Campos, 2011, Level III [39]	Descriptive study of outcomes in both primary and recurrent epithelial ovarian cancer who had been treated with optimal CRS (CC-0 or CC-1) + HIPEC	● Cytostatic paclitaxel: 60 mg/m^2^ ● In taxol allergic patients: 75 mg/m^2^ cisplatin ● 60 min at 42–43 °C	● Median operative time 380 mins ● CC-0 achieved in 38/46 patients ● Major morbidity 15.3% with ileus being most common ● Mean operative stay 6.9 days ● No mortality
Fagotti, 2012, Level III [40]	Case-control ● CRS + HIPEC (*n* = 30) ● CRS without HIPEC (*n* = 37) Platinum-sensitive recurrent epithelial ovarian cancer patients	● Oxaliplatin: 460 mg/m^2^ ● 30 min at 41.5 °C ● Systemic platinum-based chemo afterwards	Primary outcome: progression-free survival ● HIPEC: 26 months ● Non-HIPEC: 15 mos 5 year overall survival: ● HIPEC: 68.4% ● Non-HIPEC: 42.7% CC-0 reached in 96.7% Median time to systemic CT = 40 days (no diff) HIPEC toxicity in 35%, 0% 30 day mortality
Bakrin, 2012, Level III [41]	● Platinum sensitive (*n* = 184) ● Platinum resistant (*n* = 62) Multicenter prospective study of outcomes in recurrent epithelial ovarian cancer who had been treated with optimal CRS + HIPEC	● 95.5% used cisplatin alone or in combination with doxorubicin or mitomycin C ● 90 min from 44–46 °C inflow temperature	Primary outcome: grade III-IV complications ● 11.6% complications Overall median survival: 48 months in platinum-resistant and 52 months in platinum-sensitive (*p* = 0.568)
Safra, 2014, Level III [42]	Retrospective case control ● CRS + HIPEC + Systemic CT (*n* = 27) ● CRS + systemic CT only (*n* = 84) Recurrent ovarian cancer patients, stratified by BRCA gene mutation status	● Cisplatin 50 mg/m^2^ + doxorubicin 15 mg/m^2^ ● Paclitaxel 60 mg/m^2^ + carboplatin AUC 4 ● Cisplatin 25 mg/l/m^2^ + mitomycin-C 3.3 mg/l/m^2^ ● 120 min at 42.5 °C	Primary outcome: median progression free survival ● HIPEC: 15 months ● Non-HIPEC: 6 months Secondary outcome: BRCA status progression free survival ● BRCA HIPEC: 20.9 months ● BRCA non-HIPEC: 12.6 months 5 year survival: ● HIPEC: 79% ● Systemic CT: 45%
Cascales Campos, 2014, Level III [32]	Retrospective case control study before and after introducing HIPEC in 2008 ● Before (*n* =22) ● After (*n* = 32)	● Cytostatic paclitaxel: 60 mg/m^2^ ● In taxol allergic patients: 75 mg/m^2^ cisplatin ● 60 min at 42–43 °C	Primary outcome: Progression-free survival ● Before- 77% at 1 yr, 23% at 3 yr ● After- 77% at 1 yr, 45% at 3 yr Non-significant differences in PFS No significant differences in morbidity (23% before, 28% after)
Le Brun, 2014, Level III [43]	Retrospective multicenter case-control study Recurrent ovarian cancer patients ● CRS + HIPEC (*n* = 23) ● CRS + Control (*n* = 19)	● Cisplatin: 16 mg/m^2^ ● Eloxatin 6 mg/m^2^ ● Mitomycin 1 mg/m^2^ ● 42 °C ● Cisplatin 60 min, other two 30 min	Overall survival at 4 years ● HIPEC: 75.6% ● Control: 19.4% CC-0 rate 100% No mortality Mean length of stay ● HIPEC: 17 days ● Control: 11 days
Delott, 2015, Level III [44]	Retrospective single-center review looking at morbidity and survival in elderly patients (*n* = 15) Age >70 years (median age 72) Relapsed ovarian cancer with peritoneal carcinomatosis	● Cisplatin 50 mg/m^2^ + doxorubicin 15 mg/m^2^ ● 60 min at 43 °C	● Progression-free survival: 15.6 months ● Medial overall survival: 35 months ● CC-0 in 9/15 cases, CC-1 in 6/15 ● Grade 3–4 in 20% of patients ● Median length of stay: 13 days (average 20)
Spiliotis, 2015, Level I [45]	Prospective randomized controlled trial ● CRS + HIPEC + Systemic CT (*n* = 60) ● CRS + Systemic CT (*n* = 60) Recurrent stages IIIC-IV epithelial ovarian cancer	● Platinum-sensitive: cisplatin 100 mg/m^2^ + paclitaxel 175 mg/m^2^ ● Platinum-resistant: doxorubicin 35 mg/m^2^ + paclitaxel 175 mg/m^2^ or mitomycin 15 mg/m^2^ ● 60 min at 42.5 °C	Primary outcome: mean overall survival ● HIPEC: 26.7 mos ● Non-HIPEC: 13.4 mos No statistical difference between platinum-resistant and -sensitive in HIPEC group CC-0 in 65% HIPEC, 55% control CC-1 in 20% HIPEC, 33.3% control
Petrillo, 2016, Level III [46]	Retrospective review of 5 and 7 year survival outcomes with median follow up of 73 months in 70 patients Recurrent platinum-sensitive disease who had undergone CRS + HIPEC	● Oxaliplatin 460 mg/m^2^ in 43 cases for 30 min ● Cisplatin 75 mg/m^2^ in 17 cases for 60 min ● 41.5 °C ● Systemic platinum + taxane postoperatively	● CC-0 in 88.6%, CC-1 in 11.4% ● Median progression-free survival 27 months ● 5 year survival: 52.8% ● 7 year survival: 44.7% ● 30 day complication rate: 35.7% with two grade 3, and four grade 4

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
