# Peer review of "Hyperthermic Intraperitoneal Chemotherapy in Ovarian Cancer"

_diagnostics, 2020, doi:10.3390/diagnostics10010043_

Round 1
Reviewer 1 Report
Dear authors
The paper describes the use oh HIPEC in primary and recurrent ovarian cancers. Both surgical and oncological point of views were well described. The paper is clearly written and of interest for surgeons, medical oncologist and translational scientists.
There are a few minor points to be addressed:
Literature appears old with a few cited papers in the last two years.
Conclusions should be expanded with a point of view from authors on what could be the best strategies to follow for improving the use of HIPEC in clinic.
Author Response
We appreciate the recommendations provided by the first reviewer. He or she acknowledged that our review includes literature that seems dated. Currently, HIPEC use in ovarian cancer is not yet standard of care. Recent literature on this topic is scant, and one of our goals with this review article is to highlight areas for future exploration. Hopefully, our article may spark further interest into new investigation into this topic.
We expanded the conclusion to further clarify the points of interest/debate to make HIPEC in ovarian cancer more standardized and clinically applicable.

Reviewer 2 Report
This review emphasises on the use of Hyperthermic Intraperitoneal Chemotherapy (HIPEC) in Ovarian Cancer. It nicely highlights the brief history of HIPEC and its use in primary as well as recurrent cancer.
Author Response
We appreciate the review and positive feedback by this reviewer. Grammatical changes were made as indicated in the edited manuscript.

Reviewer 3 Report
This is an overview on the introduction, evolution and effectiveness of HIPEC along with CRS in the treatment of primary and recurrent ovarian cancer. 

Although comprehensive, the manuscript should be more succinct and focused and the issues related to the use of HIPEC in ovarian cancer should be more clearly presented in the last paragraph of conclusions.
Please pay also attention to some minor spelling mistakes (see lines 114,181, 182).
Author Response
We appreciate the recommendations by this reviewer. We shortened the manuscript to better clarify our points, particularly in the conclusion, but also eliminated portions to make it more succinct. The lines were numbered but spelling errors did not correspond as described by the reviewer. Made grammatical and spelling changes as needed throughout the document.
